# Decoding biomolecular condensate dynamics: an energy landscape approach

**Subhadip Biswas**[ID][1], **Davit A. Potoyan**[ID][1,2,3]*

**1** Department of Chemistry, Iowa State University, Ames, Iowa, United States of America, **2** Department of Biochemistry, Biophysics and Molecular Biology, Iowa State University, Ames, Iowa, United States of America, **3** Bioinformatics and Computational Biology Program, Iowa State University, Ames, Iowa, United States of America

* potoyan@iastate.edu

## Abstract

Many eukaryotic proteins and RNAs contain low-complexity domains (LCDs) with a strong propensity for binding and driving phase separation into biomolecular condensates. Mutations in LCDs frequently disrupt condensate dynamics, resulting in pathological transitions to solid-like states. Understanding how the molecular sequence grammar of LCDs governs condensate dynamics is essential for uncovering their biological functions and the evolutionary forces that shape these sequences. To this end, we present an energy landscape framework that operates on a continuous 'stickiness' energy scale rather than relying on an explicit alphabet-based sequence. Sequences are characterized by Wasserstein distance relative to thoroughly shuffled or random counterparts. Armed with an energy landscape framework, map diagrams of material and dynamical properties governed by key energy landscape features modulated by the degree of complexity in LCD arrangements, including the periodicity and local disorder in LCDs. Highly periodic LCD patterns promote elasticity-dominated behavior, while random sequences exhibit viscosity-dominated properties. Our results reveal that minimum sticker periodicity is crucial for maintaining fluidity in condensates, thereby avoiding transitions to glassy or solid-like states. Moreover, we demonstrate that the energy landscape framework explains the recent experimental findings on prion domains and predicts systematic alterations in condensate viscoelasticity. Our work provides a unifying perspective on the sequence-encoded material properties whereby key features of energy landscapes are conserved while sequences are variable.

## Author summary

This study investigates why proteins with low complexity domains are ubiquitous in liquid-like structures within cells called biomolecular condensates. The material properties of condensate are essential for cellular functions and for dictating diffusion and reaction of regulatory molecules. Using an energy landscape-based model, we elucidate

---

**Data availability statement:** All data are in the manuscript and/or supporting information files.

**Funding:** This work was supported by the National Institutes of Health with a grant No. R35GM138243 (to DAP). The funders had no role in study design, data collection and analysis, decision to publish, or preparation of the manuscript.

**Competing interests:** The authors have declared that no competing interests exist.

how the degree of complexity in biomolecular sequences quantified by periodicity and disorder in sticky regions impacts condensate dynamics and material properties. This work bridges the gap in understanding the relationship between sequence and material properties of biomolecular condensates, paving the way for innovations in cellular biology and disease treatment.

## Introduction

A significant fraction of the eukaryotic proteome contains sequences with low-complexity domains (LCDs) [1–3]. Proteins with LCDs display high conformational flexibility and affinity for binding and phase separation into biomolecular condensates, which engage in numerous organizational, regulatory, and signaling functions [1,4]. The material properties and dynamics of condensates appear crucial for functions and are tightly regulated in cells [5,6]. Misregulation of material properties often leads to loss of condensate fluidity and an irreversible transition to solid-like states associated with pathological diseases [7,8]. *In vitro* experiments have further demonstrated that dynamic and material properties of condensates are sensitive to sequence mutations, particularly in regions containing LCDs [9–15]. At the same time, globally, the sequences of proteins with LCDs, which often correspond to intrinsically disordered regions (IDRs), are not well-conserved [16]. Therefore, it is puzzling how sequence variability is compatible with persistent cellular function and, if so, how choice of specific sequence patterning is ultimately chosen. Some of these observations have been rationalized by sticker and spacer models, which have provided some insights into sequence-dependent structural and thermodynamic properties of condensates [17–20]. For instance, in recent reports, flow activation energies of various condensate phases were measured, finding that localized sticky motifs in sequences dramatically alter network reconfiguration times in condensates, thereby contributing to observed sequence dependence of viscosities [10,21,22]. Nevertheless, the molecular grammar and evolutionary forces behind the widespread presence of LCD patterns in condensate-forming biomolecules remain poorly understood.

For foldable proteins, the energy landscape framework has provided clear biophysical principles for sequences of natural proteins [23–25]. For protein folding, evolutionary pressure has favored sequences that minimize frustrated interactions in proteins [23–25], which in turn minimize kinetic traps, thereby ensuring rapid and robust folding to native states. Insight from energy landscape theory has revolutionized protein design and structure prediction in the last decade [26–28]. It is, therefore, intriguing to look for an appropriate framework for elucidating the features of energy landscapes of condensate-forming biomolecules. To this end, we introduce a Free Energy Landscape of Stickers (FELaS): a model for phase-separating biomolecules that employs a continuous "stickiness" scale to investigate the effects of low-complexity sequence patterns on condensate dynamics and material properties. FELaS extends the binary sticker-and-spacer model, identifying conditions under which the binary classification succeeds and fails (Fig 1).

In FELaS the biomolecular sequence is defined as "stickiness" residues relative to the lowest baseline $\delta E = \varepsilon_i - \varepsilon_{sp}$. For instance, in a typical phase-separating sequence of proteins $(YGYGG)_n$, residue $G$ will correspond to $\delta E = 0$ and residues $Y$ to the positive finite value of $\delta E > 0$. Unlike the binary sticker spacer models, there is a continuous energy scale with which one can describe all biomolecular sequences. Since we are interested in low-complexity sequence motifs, we will refer to clusters of residues with $\delta E \approx 0$ as spacers and clusters of residues with $\delta E \approx 1$ as stickers to simplify the language. We will, however, note cases where

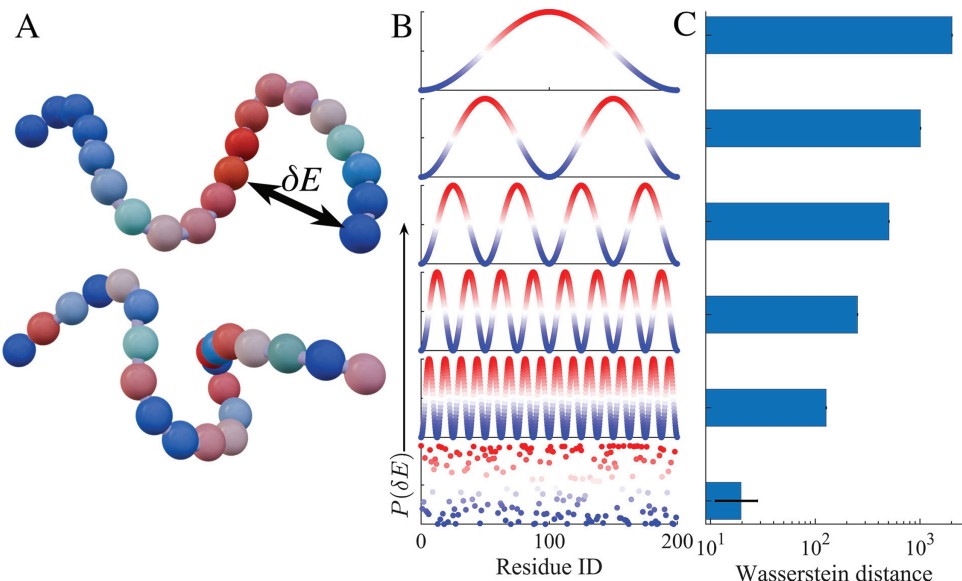

**Fig 1. (A) Schematic depiction of an alphabet-free sticker-spacer framework for protein chains, where associative interactions, $\delta E$, between domains are sampled from an energy landscape.** The blue-red color scheme represents distinct motif IDs: blue indicates spacers, while shaded blue-red regions denote stickers with varying interaction energies. (B) Schematic representation of the probability distribution of the residues' non-bonded interaction energy, $P(\delta E)$, transitioning from higher periodicity continuous sinusoidal function to systematically increasing frequencies, with randomly arranged residues shown at the bottom. (C) The Wasserstein distance of the distributions in (B) relative to the randomly distributed residues decreases as frequency increases.

such a language is not appropriate in cases where there is no clear energetic separation. Leveraging an energy landscape framework, we set out to explore the contribution of sequence complexity and periodicity of sticker repeat domains (Fig 1) in dynamic and viscoelastic properties and shed light on recent experiments probing sequence-dependent material properties of condensates [10,29].

Using FELaS, we also map a broad phase diagram of material properties driven by key energy landscape features, including the periodicity of LCDs and the disorder within LCDs. Highly periodic sequences promote elasticity-dominated behavior, while random sequences exhibit viscosity-dominated properties. Our results reveal that minimum sticker periodicity is crucial for maintaining fluidity in condensates, thereby avoiding transitions to glassy or solid-like states. Moreover, we demonstrate that the energy landscape framework aligns with recent experimental findings on prion domains and predicts systematic alterations in condensate viscoelasticity through sticker periodicity and strength variations. These insights provide a unifying perspective on the sequence-encoded material properties of biomolecular condensates, advancing our understanding of their dynamics and functional roles.

## Free Energy Landscape of Stickers (FELaS)

Here, we describe the physical, motivational, and computational details of the Free Energy Landscape of stickers (FELaS) (Fig 1). The key idea is adopting a continuous relative energy scale $\delta E$, which captures continuous sequence variation of residues' ability to associate without operating with an explicit alphabet of amino acids. Clusters with a high affinity for self-association will have $\delta E \approx 1$ and can be referred to as stickers. Clusters with low affinity for self-association will have $\delta E \approx 0$ and can be referred to as spacers. There will also be cases

where the language of stickers and spacers ceases to be helpful. Hence, FELaS is better seen as a continuous generalization of a binary sticker-spacer framework.

In FELaS the chains of biomolecules are coarse-grained into per-residue beads connected by anharmonic FENE (Finite Extensible Nonlinear Elastic) springs, described by the potential $U_{\text{FENE}}(r) = -\frac{1}{2}kR_0^2\ln(1 - r^2/R_0^2)$. Here, $r$ represents the distance between adjacent beads, with a spring coefficient of $k = 20\epsilon/\sigma^2$ and maximum extensibility of $R_0 = 1.5\sigma$, where $\sigma$ denotes the bead diameter, establishing the length scale for LJ (Lennard-Jones) simulations. Bond angles between consecutive beads are constrained by an angular harmonic potential, $U_\theta = K_\theta(\theta - \theta_0)^2$, where $K_\theta$ denotes the potential energy and $\theta_0 = 150°$ is the equilibrium angle. Non-bonded interactions are incorporated through the Lennard-Jones (LJ) potential, expressed as $U_{\text{LJ}}^{ij}(r) = 4\epsilon_{ij}[(\sigma/r)^{12} - (\sigma/r)^6]$, effective within a cutoff distance $r_c$, where $U_{\text{LJ}}(r)$ linearly goes to zero at $r_c$.

The strengths of non-bonded interactions $\epsilon_{ij}$ are calibrated to range between well depth 0 (repulsive potential) and $kT$, following a periodic modulation $\epsilon_{ii} = \sin^2(2\pi\text{Res}_i/k)$, $\text{Res}_i$ with period $k$ set at 10, 25 and 100 in Fig 2A and 2B. Inter-motif interactions $i \neq j$, where the energy between $i$ and $j$ monomers is weaker than intra-motif interactions ($i = j$), are established via $\epsilon_{ij} = \epsilon_{ii}\epsilon_{jj}$, as illustrated in the pairwise interaction energy landscape diagram in Fig 2C.

The interaction energies with the well depths $\epsilon_{ij} \leq 0.1$ are set to 0, representing WCA (Weeks-Chandler-Andersen potential) repulsive interactions. They are termed "spacers", while the beads with attractive interactions are referred to as "stickers." The functional form ensures that the total energy across different periodicities remains the same, allowing one

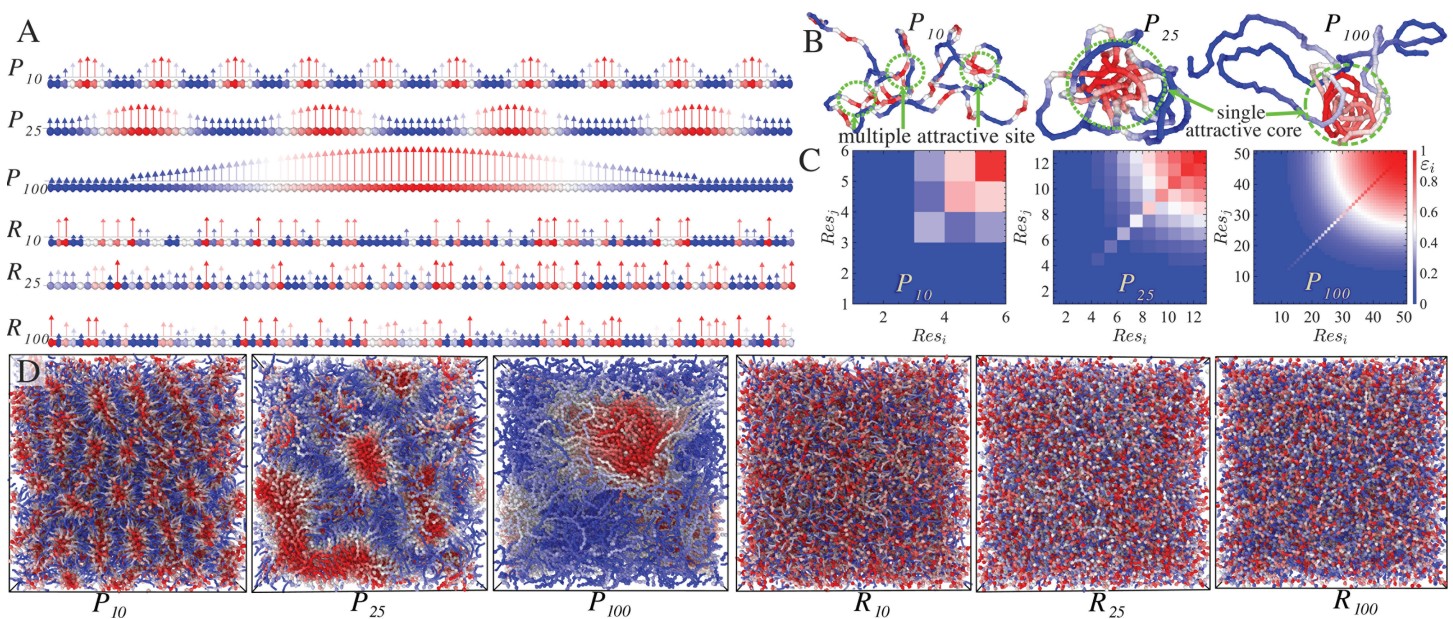

**Fig 2. Schematic description and simulation snapshots of the sequence-specific spatial arrangement of residues.** The variations of i) periodically distributed sticker-spacer motifs $P_{10}$, $P_{25}$, and $P_{100}$, and ii) randomly distributed sticker-spacer motifs along the chain backbone $R_{10}$, $R_{25}$, and $R_{100}$ are shown in (A). Conformations of the FELaS chain in very dilute solutions are depicted in (B). The interaction strengths between distinct stickers are illustrated in the pairwise interaction matrix in (C), it depicts the red motif exhibiting an interaction energy of $\epsilon = 1$, representing the highest sticker strengths. In contrast, a blue motif denotes spacers with the lowest interaction energy. (D) As the periodicity of the sticky region along the chain increases, intermediate sticker strengths are introduced, conserving total energy along the chain. Simulation snapshots of the bulk system of chains with periodic and randomly distributed stickers while fixing all other parameters. A membrane-shaped layer forms for $P_{10}$, a cluster domain for $P_{25}$, and a larger sticky cluster for $P_{100}$. The structural architecture transitions into a homogeneous condensed phase for randomly distributed $R_{10}$, $P_{25}$, and $R_{100}$.

to dissect the impact of sequence patterns on the viscoelastic and structural properties of condensates. We have considered the following sticker spacer motifs using a notation of $P_k$ denoting sequence energy landscape with sticker periodicity $k$ = 10, 25, 100 (Fig 2A). The equilibrium and dynamic behaviors of these sequences are discussed in Fig 3. We have then introduced randomness in sequence in steps, denoting $P_k^A$ where sticker motifs are randomized while fixing spacers, $P_k^B$ where alternatively higher interaction strength sticker - spacers or lower interaction strength sticker motifs are distributed along the chain (Fig 4). Taking the periodic chains and randomizing residue arrangements leads to the extreme random heteropolymers denoted as $R_k$, which we use to understand the impact of periodic sequence motifs (Fig 2A). Simulation snapshots illustrating these configurations of the bulk solution in equilibrium are presented in Fig 2D.

## Results

### Thermodynamics and dynamics of low complexity periodic motifs vs random heteropolymers

Before delving into material properties encoded by energy landscapes of stickers and spacers, we first analyze how patterns of stickers and spacers encode the thermodynamics of phase equilibrium. All the simulated chains have the same length $N_p$ = 200, mimicking the length scales of LCDs commonly found in condensate-forming proteins. We systematically vary the

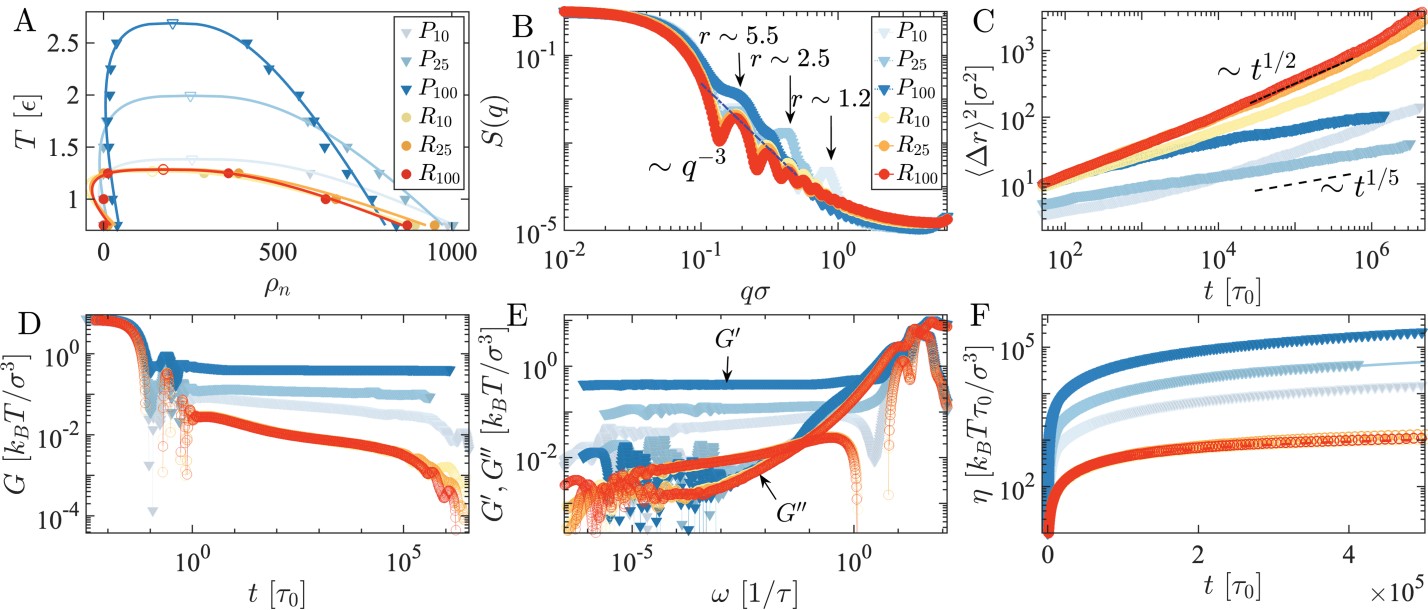

**Fig 3. Equilibrium and non-equilibrium properties of the programmable energy landscape sticker-spacer model: A) The phase diagram of periodically distributed chains (blue shaded) condensates reveals that $P_{100}$ exhibits a critical temperature $T_c$ more than two-fold higher than that of $P_{10}$ due to the lower periodic occurrence of the sticker.** Conversely, the randomly distributed (red shaded) for the same systems shows a lower critical temperature $T_c$. B) Anomalous peaks in the structure factor $S(q)$ of periodically distributed chains $P_{\text{period}}$ characterize equilibrium folded sticky domain sizes absent in randomly distributed chains. C) With increasing period, periodically distributed chains $P_{\text{period}}$ displays more glassy arrested behavior, following $t^{1/5}$, whereas randomly distributed chains follows $t^{1/2}$. D) The complex modulus $G$ of randomly distributed chains decays faster than periodically distributed chains. E) Elastic ($G'$) and viscous ($G''$) moduli exhibit a high-frequency viscous-dominated regime and an elastic-dominated crossover at an intermediate frequency range. For $\omega \to 0$, a Maxwell viscous-dominated trend is observed for randomly distributed chains, while a Kelvin-Voigt $G'$ plateau is observed for periodically distributed chains. F) periodically distributed systems are highly viscous, with viscosity increasing with the periodicity of the stickers, whereas randomly distributed systems have an order of magnitude lower viscosity.

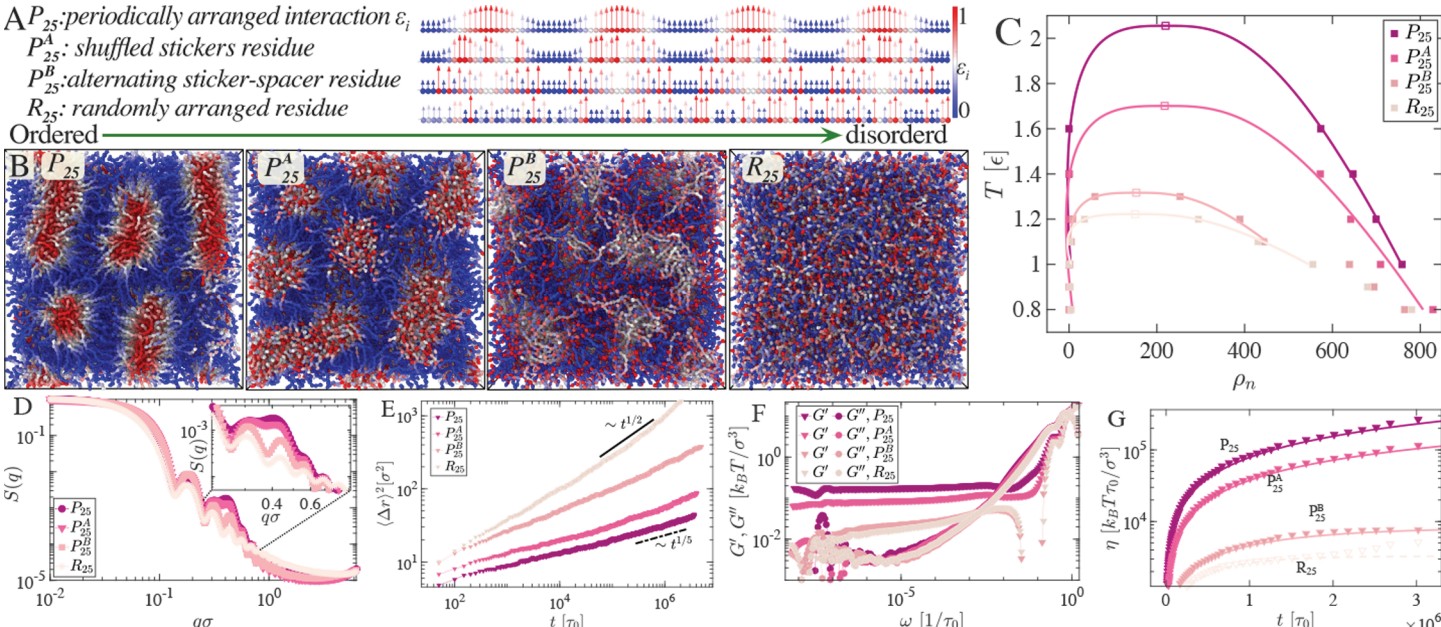

**Fig 4.** Simulation snapshots depict the transition from ordered to disordered energy landscapes of monomer interactions for various sticker and spacer arrangements. Alongside periodically distributed $P_{25}$ (A & B) and randomly distributed $R_{25}$ (A & B) chains, two new sticker-spacer arrangements are introduced. $P_{25}^A$ (A): sticker residues are shuffled while keeping spacer periodicity fixed, and its corresponding bulk simulation snapshot is shown in (B). $P_{25}^B$ (A & B): an alternative sticker-spacer chain arrangement and its corresponding bulk simulation snapshot shown here. (C) shows the phase diagram of the four different sequences pattering where $T_c$ of $P_{25}$ is almost double of $R_{25}$. (D) Static structure factors $S(q)$ for condensates formed by different sticker-spacer arrangements. (E) Mean square displacement averaged over all beads and shown for condensates formed by different sticker-spacer arrangements. (F) Elastic and viscous moduli for different sticker-spacer arrangements. (G) The viscosity was computed for the different sticker-spacer arrangements.

periodicity of the sticker motifs (where $\mathrm{Res}_{\mathrm{Period}} = 10, 25, 100$) while keeping the total "sticker energy" conserved. Despite having constant sticker energy, the critical temperatures exhibit significant variation with sticker periodicity (Fig 3A). For shorter period sequences, *i.e.*, $P_{10}$, the critical temperature is $T_c \approx 1.4\epsilon$. For longer periodic sequences like $P_{25}$, the critical temperature increases to $T_c \approx 2\epsilon$. In the case of much longer period $P_{100}$, $T_c$ is almost twice as high as that of the $P_{10}$ sequence arrangements, $T_c \approx 2.7\epsilon$ (Fig 3A).

As the extreme case, we examine the fully randomized arrangement of sticker and spacer motifs denoted as $R_{10}, R_{25}, \& R_{100}$. Intriguingly, regardless of the permutation of the periodically distributed chain, the phase diagram of the randomly arranged sequences manifests approximately the same critical temperatures, denoted as $T_c \approx 1.4\epsilon$, (Fig 3A). Notably, randomizing the spatial arrangement of the sticker and spacer motifs of $P_{100}$ results in a halving of the critical temperature from $T_c = 2.7\epsilon$ to $T_c = 1.4\epsilon$. Observing the phase diagram, one notices that periodically arranged stickers demonstrate a greater propensity for phase separation than randomly distributed stickers. Moreover, phases will likely be more stable at higher temperatures if the periodicity broadens. Augmenting the periodicity of attractive beads within our model resembles increasing the number of stickers, akin to folded binding domains, in a protein structure [17,30]. Experiments have demonstrated that mutations at sticker sites, rendering them functionally inert, reduce the tendency of proteins to undergo phase separation [31], a phenomenon analogous to our findings.

**Structure factor:** Here, we explore the correlation between sticker spacer arrangements and the emergence of local structural order. We observe an increase in the thickness of the sticker layer or domain size with elongated periodic chains, resulting in a shift of scattering

peaks towards lower $q$-values (see in (Fig 3B)). In randomly distributed chains, $S(q)$ decays as $q^{-3}$, without anomalies in the peak, while in periodically distributed chains, one peak is significantly higher, indicating local structures of sticker domains. The reciprocal of the peak position in Fourier space corresponds to the thickness layer ($r \approx 1.2\sigma$) in $P_{10}$, which increases longer periodicity for $P_{25}$ ($r \approx 2.5\sigma$), and further increases for $P_{100}$, the shift to even lower values ($r \approx 5\sigma$), as indicated in (Fig 3B). The $S(q)$ reveals the condensates' local architecture, which, as we shall see in subsequent sections, influences nonequilibrium rheological and viscoelastic properties. An increase in sticky domain size suggests the creation of folded sticky domains, enhancing stability and promoting a more viscoelastic behavior. Conversely, randomly distributed chains exhibit fluidic behavior due to the absence of local structure in the condensed phase. Equilibrium contact map calculations of condensates support the local structures in $S(q)$ peaks. Specifically, the contact maps illustrate spatial proximity ($r_c < 2\sigma$) between motifs $i$ and $j$ within a biomolecular condensate (Fig A in S1 Text). In periodically distributed chains, well-formed domains of sticker motif contacts are evident, while consistent contact between spacers is lacking. Conversely, randomly distributed chains exhibit no consistent contact regions capable of forming sticker domains. The radial distribution function $g(r)$ between sticky particles are shown in (Fig E in S1 Text), where the short-distance behavior is nearly identical irrespective of the sequence patterning.

**Mean squared displacement:** We use the mean squared displacement (MSD) to elucidate the subdiffusive and arrested dynamics of condensates arising from different sticker-spacer patterning. The MSD is defined as $\langle \Delta r^2(t) \rangle = Dt^\alpha$, where $D$ is related to the diffusion constant and $\alpha$ is the scaling exponent. Our analysis reveals subdiffusive behavior, with MSD scaling as $\langle \Delta r^2(t) \rangle \sim t^{1/2}$ to $t^{1/5}$, driven by the formation of sticky domains and caging dynamics (Fig 3C). The MSD exhibits a $t^{1/2}$ scaling for randomly organized sequences, reflecting slowed molecular motion due to topological entanglements and localized interactions. Similar subdiffusion behavior has been reported in protein droplets, where the viscoelastic nature of the droplet results in significant differences in protein diffusivity between its interior and interface [32]. This subdiffusive scaling is consistent across different periodicities of sticker molecules, even when the stickers are randomly distributed along the chain (Fig 3C). As the degree of periodicity broadens, sticky clusters grow, leading to spatiotemporally arrested condensed phases that exhibit progressively slower dynamics, as shown by the blue curves in Fig 3C.

Notably, the non-monotonic trend among the $P_n$ systems arises from the combined effects of system dynamics and spatial organization. For $P_{10}$, the sticky domains are less stable and highly dynamic, resulting in behavior similar to randomly arranged sequences at longer times. In contrast, $P_{25}$ shows an initial scaling akin to random chains but deviates and decreases over time, following a $t^{1/5}$ power law. The behavior of $P_{100}$ is particularly intriguing as it exhibits two distinct scaling regimes. Due to its long spacer regions, $P_{100}$ initially behaves like random chains for shorter distances. However, as the spacer dynamics exceed the spacer mesh size, the system becomes increasingly arrested, causing the $P_{100}$ curve to surpass those of lower $P_n$ values. This anomaly highlights the interplay between sticker periodicity and spacer fluctuations, which significantly influences the long-time dynamics of these systems. These observations underscore the complex relationship between sequence organization, dynamic arrest, and molecular diffusion in condensed phases.

**Viscoelasticity:** Recent microrheology experiments have demonstrated that the material properties of biomolecular condensates can vary widely, ranging from intricate Maxwell fluids to glassy or Kelvin-Voigt viscoelastic gels, depending on their composition and maturation times [21,33–35]. Here, we elucidate how the energy landscape of sequences composed of stickers and spacers encodes the diverse viscoelastic properties observed in experiments. We

find that randomly distributed stickers (Fig 3D) show multiple relaxation times seen by fitting the complex modulus $G$ to the generalized Maxwell model $G = \sum_i G_i \exp\left(-t/\tau_i\right)$. The complex modulus $G$ goes to zero for all three randomly distributed sticker cases at long $t$, revealing Maxwell fluid behavior. Maxwell-like behavior persists also for sticker-spacer systems with short periodicity $P_{10}$ (Fig 3D). However, broadening the period of sticker domains ($P_{25}$ & $P_{100}$) changes the material properties of condensate from Maxwell fluid to Kelvin-Voigt elastic solid behavior.

The complex modulus for $P_{25}$ & $P_{100}$ gets nearly saturated at long time scales (Fig 3D). The Fourier transform of $P_{25}$ & $P_{100}$ shows that the elastic modulus $G'$ is independent of imposed deformation frequency $\omega$ at long times (Fig 3E) a signature of Kelvin-Voigt solid nature of the condensed phase. As sticker domain size increases, entangled sticker segments fold back to make a strongly elastic-dominated viscoelastic condensed phase, where $G'$ of $P_{100}$ is an order of magnitude larger relative to its randomly distributed $R_{100}$ sequence. Note that, even though the elastic dominance differs for different arrangements of the sticker and spacers, viscous response, $G''$, collapses on the same curve for all systems. The scaling behavior of Maxwell fluid in the viscous dominated long time scale regime indicates that as $\omega \to 0$, exhibits $G' \sim \omega^2$ and $G'' \sim \omega$. Thus, one can think of the aging of biomolecular condensates as rearranging the sticker residue for a long time, which makes the systems fluidic to an elastic-dominated Kelvin-Voigt nature [19]. Our simulation reveals a strong correlation between the sticker profile and bulk viscosity of condensates (Fig 3F), linked across scales using Rouse theory [36]. The periodic arrangement of the stickers results in an order of magnitude more viscous (Fig 3F) phase compared to their random counterparts. This is consistent with its thermodynamically more stable phases and structurally larger sticky domains formed by chains with periodically arranged stickers.

## Dynamical impact of local randomness in periodic stickers and spacers motifs

Having discussed the two extreme limits of periodic and random sticker patterns, we now turn to examine two intermediate cases of the disorder in stickers and spacers denoted as $P_A$ and $P_B$ (Fig 4A and 4B) respectively. We use the 25-period residue sequence as the reference for comparison. In the case of $P_{25}^A$, we fix the spacers defined by the lowest $\varepsilon$ interaction strengths and randomize sticky domains defined by the higher values of $\varepsilon$. In case $P_{25}^B$, the sticker period is fixed, but the strengths of residues in the sticker domain alternate between higher and lower values sequentially (Fig 4A). This corresponds to typical *RGRG*, *YGYG* motifs in many phase-separating proteins [1,37,38].

Simulations show that one can have very distinct bulk equilibrium structures and material properties by simply tuning the degree of order in sticker and spacer domains (Fig 4B). Chains with periodically arranged ordered sticky regions self-assembled make micelle-like architecture where the bulk is heterogeneously distributed with the sticker types (Fig 4A and 4B). A disordered sticker arrangement makes a structure-less, unstable, homogeneous melt (Fig 4B). In the intermediate states, $P_{25}^A$ & $P_{25}^B$, there are also structures present; however, they are not as prominent as $P_{25}$ (Fig 4A and 4B).

The phase diagram of these different sticker arrangements (Fig 4C) shows that extreme ordered $P_{25}$ and random arrangement of stickers possess the most stable and least stable phases, respectively. The $P_{25}^A$ & $P_{25}^B$ sequences lie in the intermediate stability regime. To structurally distinguish the architecture of these condensates, we calculate the structure factor $S(q)$. An anomalous peak in the case of $P_{25}$ & $P_{25}^A$ indicates the existence of local structure in

the condensed phase. The peak position at $q$ refers to the size of the sticker micelle-like local structures (Fig 4D).

The prominent peak is lowered in the case of $P_{25}^B$, suggesting the local structure is fading out. In the case of a fully randomized sequence $R_{25}$, one ends up with a homogeneously distributed melt system. The emergence of structural order is also reflected in subdiffusive dynamics. The exponent of the MSD decays with periodically arranged stickers $\langle \Delta r^2 \rangle \sim t^{1/5}$ (Fig 4E).

In terms of viscoelastic properties quantified by $G'$, $G''$ (Fig 4F), we find that local structural order leads to elevated elastic modulus (Fig 4F). We plotted viscosity over time, whereas the saturation value as $G^* \to 0, \eta \to \eta_0$ describes the total viscosity of the system. In condensates with periodically arranged stickers, $P_{25}$ condensates form the most viscous systems, whereas it is lowest for $R_{25}$ (Fig 4G).

One can also look into the contact time autocorrelation function, $C(t)$, which tracks the dynamics of contact formation and offers insights into the viscoelasticity [39]. Short timescales show rapid decay in $C(t)$, suggesting fluid-like behavior with frequent contact disruptions and reforms. On longer timescales, $C(t)$ indicates a shift toward a gel-like arrested state, where sustained interactions enhance viscosity and structural rigidity, mainly due to sticker-sticker interactions (Fig D in S1 Text). Periodic sticker arrangements result in longer correlation times in $C(t)$, while higher temperatures accelerate correlation decay, indicating reduced viscoelastic stabilization from sticker interactions (Fig D in S1 Text).

## Comparison with experiments

We have reported that viscoelasticity grows as a function of the periodicity of stickers and that mutations in sticker regions have the most dramatic impact on material properties. In this section, we show that these results fall within the expected range of experimentally characterized material properties despite the limited data available in the literature on the viscoelastic properties of protein condensates with varying sticker and spacer residues [21,40] (sequences are listed in the SI section D). We define periodicity for a real protein sequence to facilitate comparison with our energy landscape predictions. For this, we use a correlation length scale where the maximum energy strengths as defined by hydropathy index [41] $\lambda_{max}$, decays to half-width total maxima along the chain, $\langle \kappa \rangle$. We plot the inverse loss tangent, $1/\tan\delta$, or the ratio of $G'/G''$, as a function of $\langle \kappa \rangle \sum \lambda_i / \lambda_{max} / N_p$, where the chain length is $N_p$ and $\lambda_i$ is the energy amplitude of the $i$-th residue, and $\lambda_{max}$ is the maximum energy among the sequence residues. Note that most of the experiments have been conducted with changing residues; therefore, along with the periodicity $\langle \kappa \rangle$, we also incorporate the unitless interaction energy parameter $\lambda$. A similar increasing trend in the elastic modulus with increasing periodicity is observed in our simulation data (Fig 5B). It is consistent with the limited experimental data from [21] (Fig 5A). An alternative parametrization of these sequences using the Wasserstein distance, $W_1$, reveals similar behavior, as shown in SI (Fig F in S1 Text). This analysis underscores the predictive capability of the sequence arrangements, with the Wasserstein distance serving as a key metric to quantify deviations from randomness. Observing the viscoelastic properties of such predictive sequence patterning would provide valuable insights into the spatial molecular grammar and its relationship to material properties.

## Orientational order and correlations

A semi-flexible homopolymer chain's persistence length ($\ell_p$) exhibits resistance to bending, which characterizes the mechanical and viscoelastic properties of constitutive bulk systems. The angle between a tangent vector $\hat{t}_i$ at position $i$ and another tangent vector $\hat{t}_j$ at a distance

along the polymer contour, $s$, determines the correlation $\langle \hat{t}_i \cdot \hat{t}_j \rangle \approx e^{-\frac{s}{l_p}}$, indicating the extent (*e.g.*, persistence length, $\ell_p$) over which correlations along the contour approach zero. Here, $\langle \cdots \rangle$ is taken over all possible configurations of the polymer and ensemble of the multiple chains of specific bead type. Typically, the homopolymer with bending rigidity (or persistence length $L > \ell_p > \sigma$) reveals viscoelastic properties. As a reference, we note that tangent correlation (exponential decay) along the chain contour is spatially invariant in the case of homopolymer, as every bead is identical.

In our model, even though the chain is not homopolymeric, we find a single exponential decay tangent correlation for random sequences as depicted in Fig 6D–6F. Interestingly,

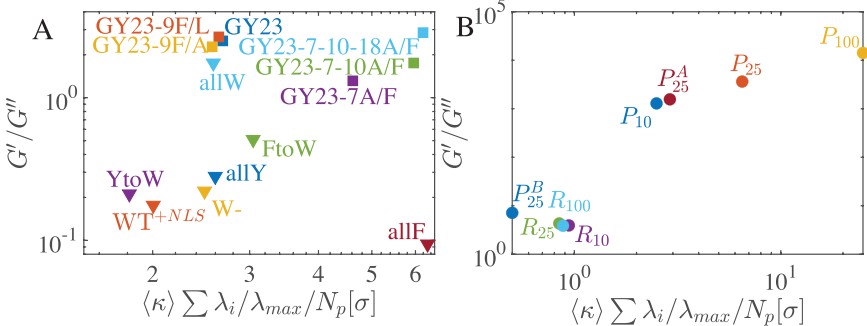

**Fig 5. Comparison of the experimental results of $G'/G''$ reported in [21] and [40] in (A) with our model of periodic and random sequences in (B) as a function of periodicity of the sequence.** Both experimental and simulation results show an increase in viscoelasticity as the periodicity of the arranged sequences increases.

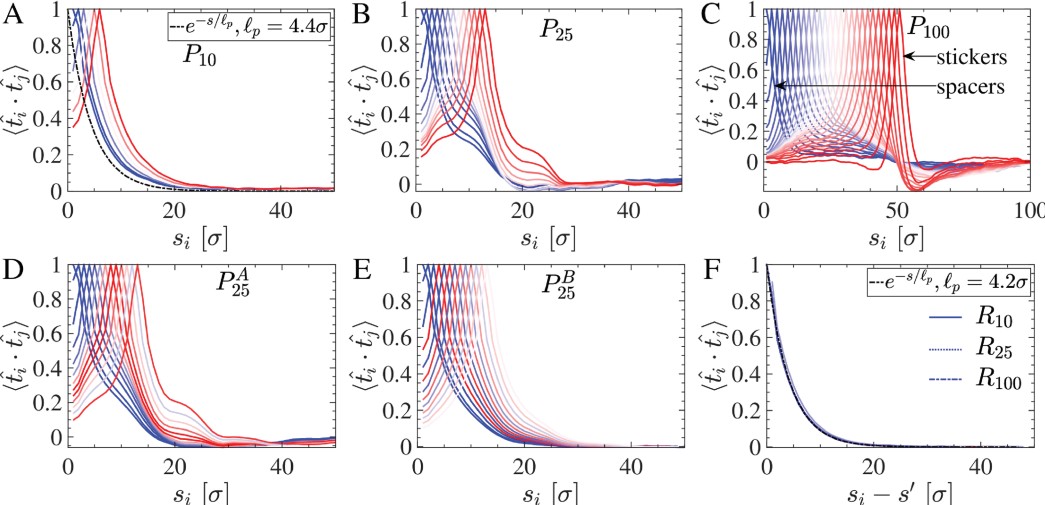

**Fig 6. Tangent-tangent correlation in condensates was calculated for the periodically distributed (A–E) and randomly distributed (F) stickers.** (A) For $P_{10}$, periodic sticky regions aggregate; however, the chains show uniform exponential decay correlation. (B) $P_{25}$ shows anomalous exponential decay in $\langle \hat{t}_i \cdot \hat{t}_j \rangle$, indicating a folded structure. (C) $P_{100}$ displays diverse position-dependent residue characteristics and long-range order; spacer residues show exponential decay, while spool-like folded domains show non-exponential decay and negative correlation. (D) $P_{25}^A$ also shows anomalous exponential decay, and (E) $P_{25}^B$ has distinct exponential decay lengths. (F) A collapsed exponential decay in the tangent correlation of randomly distributed chains ($R_{10}$, $R_{25}$, $R_{100}$) exhibits worm-like chain behavior with a persistence length of $\ell_p = 4.2\sigma$.

sticky residues exhibit different correlation characteristics in the case of periodic sticker-spacer motifs. Due to the formation of clusters among sticky residues, there is strong anti-correlation behavior [42–44]. The orientational ordering in periodically arranged sticker systems reveals that certain beads show non-monotonous decay in tangent correlation. For spacer beads with repulsive interactions, tangent correlation decays exponentially. One sees long-distance ordering as a function of sticker interaction strength, which also leads to negative correlations, dictating that chains are folding back. The strength of inter-action among sticker residues does not solely dictate the arrangement of folded sections (Fig 6). Still, it is also influenced by the spatial distribution of the sticker beads (Fig 6B and 6C).

The tangent correlation for short periodic sticker ($P_{10}$) systems exponentially decays, revealing non folded sticky patch region (Fig 6A). Therefore, these systems do not form tightly connected sticky domains and exhibit viscous-dominated rheological behavior under applied deformation. Consequently, they are also less stable at high temperatures, leading to a lower critical temperature ($T_c$) in the phase diagram than systems with broadly dis-tributed periodic sticker beads. Systematic increase in periodicity of the sticker in Fig 6B and 6C, sticker residues with $0.5 \lesssim \epsilon_{ij} < 1$ have a higher tendency to fold back compared to spacers $\epsilon_{ij} \approx 0$. Those have $\epsilon_{ij} \approx 1$. The radius of curvature of the coiled folded sticky regions can be extracted from the minimum correlation distance. Spacer residues exhibit the expected exponential decay characteristics (Fig 6C); however, with an incremental increase in sticker strength, residues tend to accumulate within a sticky domain in a folded configuration.

Periodically arranged sticker molecules show arrested dynamics, whereas the randomly distributed stickers are more fluid. We calculate the self-van Hove function to differen-tiate the spatial-temporal characteristics that resemble the distinct sub-diffusive nature (different exponent) of the dynamics. The self-part of the van Hove function $G_s(r,t) = \frac{1}{N}\langle \sum_{i=1}^{N} \delta[r - (r_i(t) - r_i(0))]\rangle$ (where $N$ is the total number of beads, $\delta$ is Dirac delta and ensemble average $\langle...\rangle$ over multiple trajectories), captures the probability distribution of par-ticle displacements over a time interval $t$ [45]. The approximation of the self-van Hove func-tion is Gaussian $G_s(r,t) \propto e^{-r^2/W(t)}$ [46]. Unlike the Gaussian $G_s(r,t)$ observed in structurally simple fluids [46], our analysis reveals nuanced characteristics, particularly in randomly dis-tributed frustrated sticker-spacer condensed phase. Frustration arises because the system cannot simultaneously minimize all interaction energies due to geometric or spatial con-straints. In this phase, exponential decay tails signify anomalous behavior [44,47,48]. In the case of periodically arranged systems, the self-van Hove function is Gaussian in nature, in early (dashed line Gaussian fit) and as late as $t \sim 10^8\tau$ (Fig 7A, 7B and 7C). Chains $P_{25}^A$, with shuffled stickers, remain Gaussian. However, $P_{25}^B$ chains, with a sticker periodicity of 25 and alternating sticker-spacer arrangements, deviate from Gaussian behavior at long times. On the other hand, even though initially randomly arranged stickers show Gaussian nature, for a long time, they fail to exhibit Gaussian behavior (Fig 7F). The frustration leads to a disordered structure within the condensed phase, significantly affecting the dynamical behavior, as evi-dent from the deviations of the Gaussian behavior. Therefore, we conclude that sticker beads are localized in the case of periodic stickers, as shown in Fig 7A–7E, compared to the ran-domly placed sticker-spacer arrangement. This temporal evolution of $G_s(r,t)$ provides invalu-able insights into the evolving microenvironment surrounding each sticky core, revealing structural rearrangements within the cluster [45].

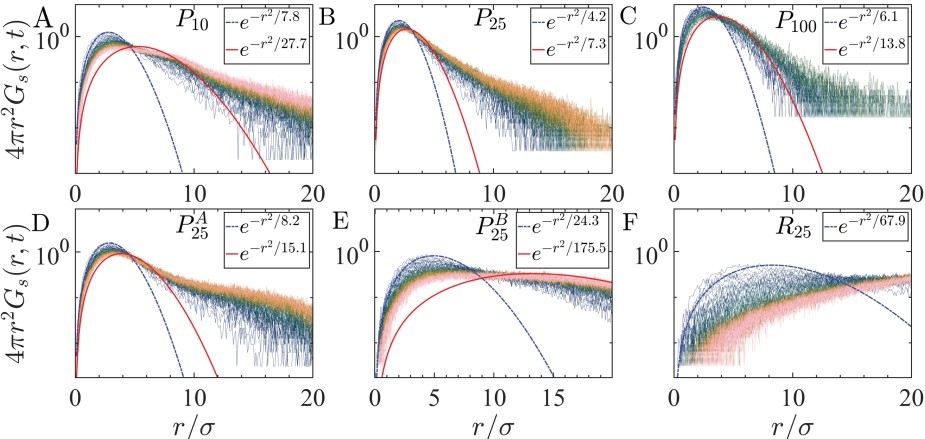

**Fig 7. The self part of the van Hove distribution function of the sticker residues, $G_s(r, t)$, are shown with the color scheme indicating the temporal evolution: dark colors represent the earlier distribution ($t \sim 10\tau$), whereas light colors depict the late-time behavior ($t \sim 10^6\tau$).** All the curves correspond to distributions taken at the same time intervals. Gaussian fits are shown for the early time ($t \sim 10\tau$, blue dashed line) and late time ($t \sim 10^6\tau$, red solid line). (A–C) represent chains with periodic sticker patterns having periods 10, 25, and 100, which remain Gaussian at long times. (D) shows $P_{25}^A$ chains with a sticker periodicity of 25 and shuffled stickers, which also remain Gaussian at long time. However, (E) depicts $P_{25}^B$ chains with a sticker periodicity of 25 and alternating sticker-spacer arrangements, which deviate from Gaussian behavior. (F) corresponds to $R_{25}$ (other $R_i$ shows similar behavior), chains with randomly arranged beads, which fail to exhibit Gaussian behavior at long times, indicating the lack of spatial organization in the system.

## Density dependent condensate architectures:

The energy landscapes of stickers encode the final equilibrium structural and dynamical properties and impact the time scales and percolation characteristics during the formation of condensates in the dilute biomolecular regime. We carried out simulations in the dilute regime observing a network-like fluid structure [29], whereas, in the dense phase, sticky domains form percolated membrane-like structures (Fig B in S1 Text). In the intermediate state, where chain densities are low, spherical micelles form, and at higher densities, micelles are no longer spherical but elongated along an axis. As density increases, these elongated tube-like micelles form large branch micelles. Above a critical density, these branched micelles connect and create a single percolated porous foam-shaped membrane-like architecture.

To quantitatively characterize the shapes of the sticky clusters, we identify the clusters and calculate the gyration tensor $\mathcal{Q}_{\alpha\beta}$ of the cluster as a function of chain concentrations $\rho$. The principal components of the gyration tensor, $R_{ii}^g$ ($i = x, y, z$, denoted by three distinct markers), are plotted as a function of $\rho$, for chain lengths $N_p = 25, 50, 100$ (Fig 8G). The vertical line separates the percolation and disperse clusters. Interestingly, all the orthogonal components of $R_{ii}^g$ are equivalent at lower densities. However, $R_{ii}^g$ degenerates as density increases, indicating that the three principal components $R_{xx}^g$, $R_{yy}^g$, and $R_{zz}^g$ which are equal for spherical clusters, become unequal. This deviation suggests that the clusters lose their spherical symmetry and transition to an elongated shape. This behavior persists across different chain lengths, as shown in Fig 8G.

The average number of particles in each cluster $n_d$ as a function of $\rho$ (Fig 8H) for different $N_p$ shows similar behavior. It is also interesting to examine the number of clusters in the system as a function of chain density, which linearly increases with density (Fig 8I and Fig C in

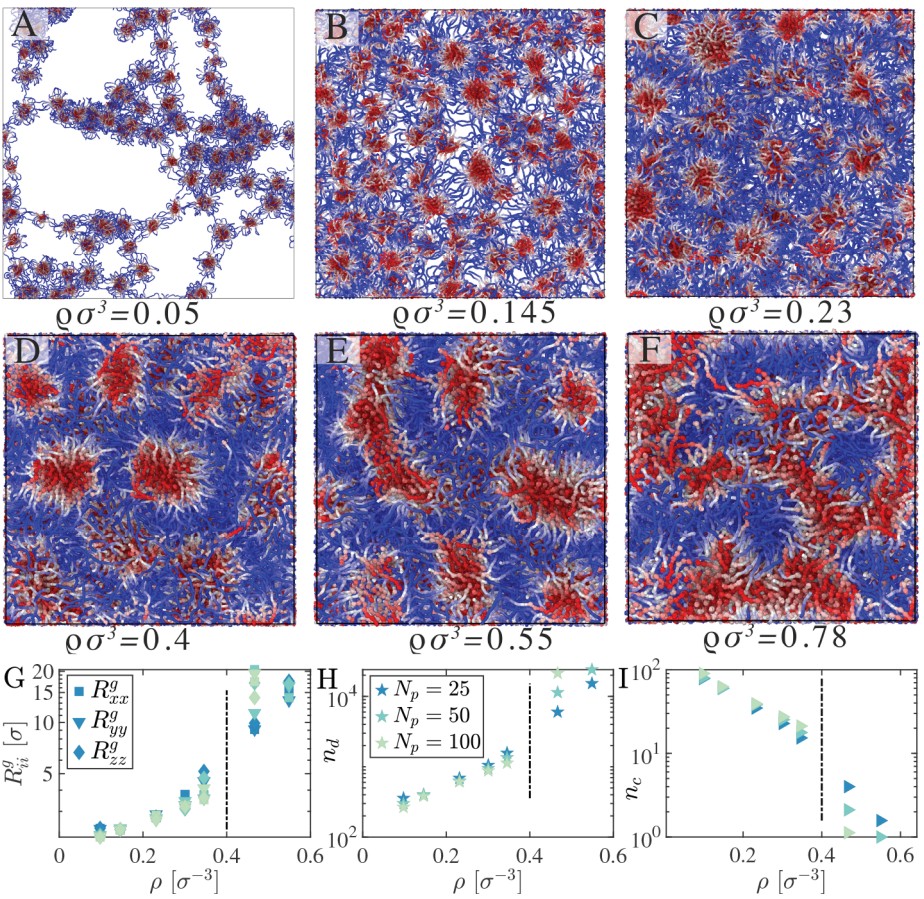

**Fig 8. Biomolecular condensate phases of $P_{25}$ at different number densities $\rho$ (by varying the box volume $L^3$) creates a range of architectures, from a network fluid structure to clusters of micelles.** (G) Orthogonal radius of gyration of sticky clusters $R_{ii}^g$, (H) average number of motifs in a single cluster, $n_d$, and (I) the average number of sticky clusters in bulk, $n_c$, as a function of box dimensions $\rho$ are shown here. The vertical lines separate the percolated single and multiple clusters in the bulk phase.

S1 Text), irrespective of the chain length. The critical length scale here is the size of the cluster or the width of the membrane-like percolated surface shape, which can be quantified by identifying the anomalous peak in $S(q)$.

To explore the morphology of clusters we use shape characteristics such as asphericity, $\Delta = \frac{3}{2} \frac{\text{Tr}\,\widehat{\mathcal{Q}}^2}{(\text{Tr}\,\mathcal{Q})^2}$ and nature of asphericity, $\Xi = \frac{4\text{Det}\,\widehat{\mathcal{Q}}}{\left(\frac{2}{3}\text{Tr}\,\mathcal{Q}^2\right)^{\frac{3}{2}}}$ where $\widehat{\mathcal{Q}}$ is the gyration tensor [49–52]. The parameter $2\sqrt{\Delta}$, ranging from 0 to 2, characterizes cluster deviation from spherical to elongated shapes, with extreme values representing spheres and rigid rods, respectively. Whereas $\cos^{-1}\Xi$ on the $x$–axis provides a quantitative measure of shape anisotropy, it delineates the distinction between oblate and prolate shapes. We report distinct micelle-like sticky cluster shapes (Fig 9) for two different scenarios: i) four different chain lengths $N_p$ (Fig 9A–9D) ii) different chain densities $\rho$. Notably, the red line delineates the structural diversity within clusters between accessible and inaccessible closed shapes. We vary the density $\rho$ for each chain length $N_p$, indicating that spherical sticky clusters form at lower densities ($\rho \leq 0.2$). In contrast, the sticky clusters form elongated micellar structures at higher densities (Fig 9). The

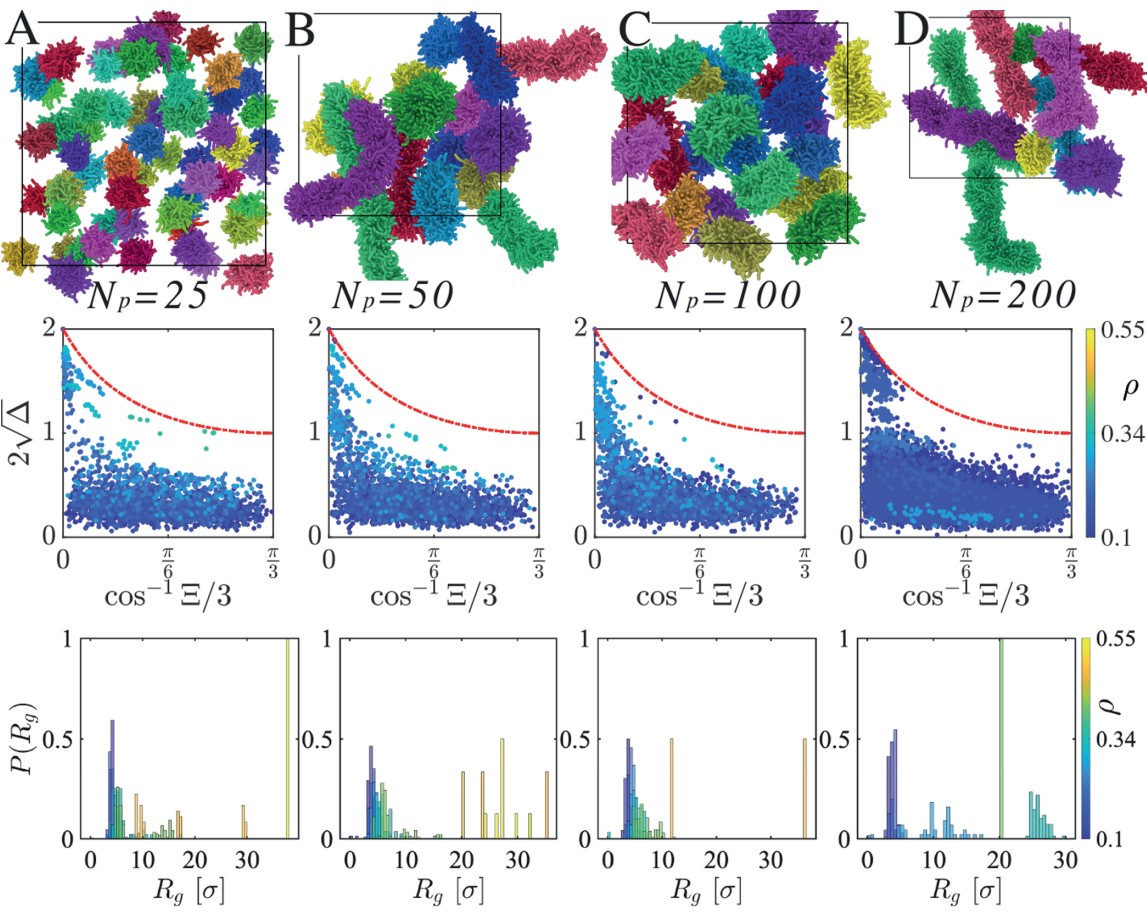

**Fig 9. The first row displays snapshots of sticker clusters for four different chain lengths *N*: (A) 25, (B) 50, (C) 100, and (D) 200, with individual clusters depicted in various colors.** In the second row, corresponding systems for each chain length illustrate how cluster shapes evolve across different density regimes, transitioning from spherical to elongated micelles. The third row presents the probability distribution of the radius of gyration, $P(R_g)$, for the sticker clusters.

histogram of the radius of gyration $R_g$ of these sticky clusters further supports this observation (Fig 9). Under dilute conditions, equilibrium clusters are smaller, exhibit more fluid-like behavior, and are loosely connected within the network (Fig D in S1 Text). Conversely, the cluster node size increases at higher densities, and the sticky nodes are tightly connected, resulting in a more elastic network.

## Discussion

Many eukaryotic proteins are characterized by extensive stretches of low-complexity domains (LCDs). However, the functional roles and evolutionary origins of LCDs, particularly those that encode conformationally flexible biomolecules, remain poorly understood. [2]. An essential functional role of LCDs appears to lie in their ability to facilitate promiscuous binding and drive phase separation into biomolecular condensates [53–55]. Despite their functional significance, LCDs exhibit low conservation, as revealed by sequence alignment analyses. Moreover, recent evidence suggests that specific binding motifs can be replaced by compensatory mutations within sequences [16]. Although the sequences of phase-separating biomolecules

can vary, studies have shown that their composition is not random. Instead, it follows specific enrichment patterns that promote the fluidity and functional dynamics of condensates [5, 6]. Misregulation of material properties of condensates leads to loss of fluidity and an irreversible transition to solid-like states associated with pathological diseases [7,8]. Computational models have provided further insights into how low-complexity sticky domains and the local rigidity of LCDs contribute to the loss of ergodic dynamics and the transition to glassy states [19,56].

Thus, decoding how sticky LCD patterns encode the material properties of condensates promises to shed light on the dynamic behavior of condensates, which are intimately linked to their sequence patterns and cellular functions [4,57,58]. To understand the puzzling paradoxes of sequence variability consistent with robust functions, we devised a Free Energy Landscape of Stickers (FELaS): a framework for exploring LCD patterns that may be evolutionarily constrained to possess both short-term material properties and long-term stability required for cellular functions. This framework provides a unifying perspective on the sequence-encoded material properties whereby key features of energy landscapes are conserved while sequences are variable. Using FELaS, we can dissect how the complexity of LCD patterns quantified by periodicity and disorder in sticky LCD regions impact the dynamics and material properties of resulting condensates. By employing Wasserstein distance, we continuously vary the complexity of the sequence and generate LCD patterns from fully periodic to random sequences.

We demonstrate that variation of a handful of energy landscape features can capture viscoelastic trends in recent experiments on prion domains [10,21,40]. We thus show that one can systematically program protein condensate viscoelasticity by altering the periodicity and stickiness of LCDs. We find periodic repeats of low-complexity sticker regions display local structure and inhomogeneities in the bulk system. In contrast, random sequences show homogeneous unstructured bulk systems, which are more viscous, as demonstrated by an order of magnitude difference in viscous moduli compared to the sequences with more periodic patterns of stickers. Phase diagrams and dynamical properties also corroborate the rheological trends; in the case of periodic LCDs, elastic response and viscosity get higher and become more subdiffusive with an increasing periodicity of the sticker motifs.

Structurally, interactions within periodic sticky LCD sequences result in a distinct oscillatory exponential decay of tangent correlations. Self-van Hove analysis reveals that a high degree of LCD periodicity can drive the system toward glassy arrested states. By varying system densities, we also observed that spherical micelle-like structures form at low densities. At higher densities, these structures transition into elongated micelles, further evolving into branched, percolated micelles at even greater concentrations.

These structural and dynamical implications of LCD patterns are likely key factors influencing the evolutionary development of phase-separating biomolecules. Overall, our study addresses a critical gap by elucidating the role of sequence complexity in governing the dynamics and material properties of biomolecular condensates. We anticipate that experiments guided by energy landscape considerations will enable more targeted mutations to systematically alter landscape characteristics, providing direct insights into the relationship between biomolecular sequences and condensate dynamics.

## Simulation methods

Molecular dynamics simulations were conducted using the LAMMPS package. The equations of motion integrated via the NVT and Langevin thermostat employing a time step of $0.001\tau$, where $\tau = \sqrt{m\sigma^2/\varepsilon}$, represents the derived unit of time and m denotes the mass of a

bead. Langevin thermostat was used with a friction factor of $0.1 m/\tau$. Chains of fixed length $N_p = 200$ were initially positioned randomly within a periodic cubic simulation box with edge lengths of $L_x = L_y = L_z = 50\sigma$. To compute the phase diagram, the simulation box was expanded along the $x$–axis by a factor of 5 (resulting in $L_x = 250\sigma$), yielding a slab of condensed phase. The phase diagram was calculated based on the density profile of the slab. The critical point was estimated via extrapolation using rectilinear diameters and the universal scaling of coexistence densities (Fig 3A), approaching asymptotically close to the critical point, expressed as $\frac{\rho_l + \rho_v}{2} = \rho_c + A(T_c - T)$ and $\Delta\rho = \Delta\rho_0 + (1 - T/T_c)^\beta$, with the exponent $\beta = 0.325$.

We equilibrated the systems inside a cubic box $L = 50\sigma$ for viscoelasticity calculations. The Green-Kubo approach has been implemented to extract viscoelastic properties. We used the multi-tau correlator method to reduce noise while preserving accurate relaxation dynamics effectively [59,60]. From the equilibrium stress autocorrelation function of an isotropic system, one can approximate complex modulus as:

$$G(t) = \frac{V}{5k_B T} \left[ \langle \Sigma_{xy}(0)\Sigma_{xy}(t) \rangle + \langle \Sigma_{yz}(0)\Sigma_{yz}(t) \rangle \right.$$
$$+ \langle \Sigma_{xz}(0)\Sigma_{xz}(t) \rangle + \frac{1}{6} \left( \langle N_{xy}(0)N_{xy}(t) \rangle \right.$$
$$\left. \left. + \langle N_{yz}(0)N_{yz}(t) \rangle + \langle N_{xz}(0)N_{xz}(t) \rangle \right) \right],$$

where $\Sigma_{ij}$ represents the off-diagonal components of the stress tensor, and $N_{ij} = \Sigma_{ii} - \Sigma_{jj}$ denotes the normal stress tensor difference. Complex viscosity, denoted as $\eta$, is calculated using $\eta = \int_0^\infty G(t')\,dt'$, where the complex modulus $G$ is defined as $G = G' + iG''$ [61,62]. For viscoelastic fluids, the real part of the complex modulus, $G'$, the elastic modulus, also known as the storage modulus or in-phase modulus, quantifies the solid-like response within the system. Whereas the imaginary part, $G''$, the viscous modulus, termed the loss modulus or out-of-phase modulus, is associated with energy dissipation through viscous flow. Fourier transform of the relaxation time dependent complex modulus $G(t) = \sum_i^{n_{modes}} G_i e^{-t/\tau_i}$, gives frequency dependent generalized Maxwell mode via $G' = \sum_i G'_i \frac{\omega_i^2 \tau_i^2}{1 + \omega_i^2 \tau_i^2}$, $G'' = \sum_i G''_i \frac{\omega_i \tau_i}{1 + \omega_i^2 \tau_i^2}$. The viscosity of viscoelastic systems can be determined as $\eta = \lim_{\omega \to 0} \frac{G''}{\omega}$.

## Supporting information

**S1 Text.** Supplementary text and Supplementary Figures.
(PDF)

## Author contributions

**Conceptualization:** Subhadip Biswas, Davit A. Potoyan.

**Data curation:** Subhadip Biswas.

**Formal analysis:** Subhadip Biswas.

**Funding acquisition:** Davit A. Potoyan.

**Software:** Subhadip Biswas.

**Supervision:** Davit A. Potoyan.

**Visualization:** Subhadip Biswas.

**Writing – original draft:** Subhadip Biswas, Davit A. Potoyan.

**Writing – review & editing:** Subhadip Biswas, Davit A. Potoyan.

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
