## [Decision Letter · Decision Letter 0]

25 Nov 2024

PCOMPBIOL-D-24-01736Decoding Biomolecular Condensate Dynamics: An Energy Landscape ApproachPLOS Computational Biology Dear Dr. Potoyan, Thank you for submitting your manuscript to PLOS Computational Biology. The reviewers were overall very enthusiastic about your manuscript but suggested some changes for clarification and to further improve impact. Therefore, we invite you to submit a revised version of the manuscript that addresses the points raised during the review process. Please submit your revised manuscript within 30 days Jan 24 2025 11:59PM. If you will need more time than this to complete your revisions, please reply to this message or contact the journal office at ploscompbiol@plos.org. Please include the following items when submitting your revised manuscript: * A response letter that responds to each point raised by the editor and reviewer(s). You should upload this letter as a separate file labeled 'Response to Reviewers'. This file does not need to include responses to formatting updates and technical items listed in the 'Journal Requirements' section below. * A marked-up copy of your manuscript that highlights changes made to the original version. You should upload this as a separate file labeled 'Revised Manuscript with Track Changes'. * An unmarked version of your revised paper without tracked changes. You should upload this as a separate file labeled 'Manuscript'. If you would like to make changes to your financial disclosure, competing interests statement, or data availability statement, please make these updates within the submission form at the time of resubmission. Guidelines for resubmitting your figure files are available below the reviewer comments at the end of this letter. We look forward to receiving your revised manuscript. Kind regards, Peter M KassonAcademic EditorPLOS Computational Biology Arne ElofssonSection EditorPLOS Computational Biology

Feilim Mac Gabhann

Editor-in-Chief

PLOS Computational Biology

Jason Papin

Editor-in-Chief

PLOS Computational Biology

**Journal Requirements:**

At this stage, the following Authors/Authors require contributions: Subhadip Biswas, and Davit A Potoyan. Please ensure that the full contributions of each author are acknowledged in the "Add/Edit/Remove Authors" section of our submission form.

5) Please provide a complete Data Availability Statement in the submission form, ensuring you include all necessary access information or a reason for why you are unable to make your data freely accessible. If your research concerns only data provided within your submission, please write "All data are in the manuscript and/or supporting information files" as your Data Availability Statement.

2) State what role the funders took in the study. If the funders had no role in your study, please state: "The funders had no role in study design, data collection and analysis, decision to publish, or preparation of the manuscript.".

**Reviewers' comments:**Reviewer's Responses to Questions

Reviewer #1: The authors provide a minimal model to characterize the equilibrium and non-equilibrium behaviors of ‘protein’ condensates. Instead of considering the exact identity of the amino acids for instance they consider the relative binding strength of interaction sites and define these binding interactions in a continuous manner – yielding an ‘energy landscape’ of interactions that can be arranged in a periodic fashion to decode the role on thermodynamic and transport properties of condensates. The manuscript is very well-written, the analyses are rigorous and the paper features quite high-quality figures/images. There are several coarse-grained models and sticker-spacer type models that have been developed and used to interrogate similar questions. Where this approach is different is in the generality of how it can be used to investigate how properties of condensates are encoded by their sequences.

The paper is already at a high standard, but there are some areas where the narrative can be improved/where details are needed to fully appreciate that value of this contribution/reproduce the work.

1. The authors should state upfront clearly what they mean by alphabet-free. This language is somewhat abstract.

2. However, use of an ‘alphabet-free’ model in probing these properties is appealing, as it allows for generality but is still detailed enough to capture effects of patterning and composition. However, the constant use of the term “sticker-spacer” throughout the paper defeats the purpose of an ‘alphabet-free’ approach / seems contradictory. The fact is it is almost impossible to delineate protein residues as pure stickers or spacers, which is exactly why their model is a good way of addressing this. But then in calling things stickers and spacers the authors are making assignments that are not needed. It also causes one to think of this problem in a binary way, which is not what they are doing or intend to do. It would be beneficial to get rid of this language and state clearly why their model is a better way at approaching this problem.

3. The abstract includes a lot of field-specific language that would make the work less accessible to abroad audience. A key message is that these interactions exist on a continuum and that the context matters. Making it clear what this model offers, how it differs from other coarse-grained approaches (of which there are many), and what is the value in approaching the problem in the way they do would is needed. What is the benefit of this approach versus use assigning a hydropathy score to residues?

4. The introduction stresses connections to energy landscape theory but this does not pay off upon reading the paper. In the discussions connecting back to the idea that the underlying sequence encodes the thermodynamics and dynamics of condensates in a similar way in which it encodes behavior of single proteins would be powerful in driving their message.

5. How do they envision others using this approach? Is there a set prescription for mapping real proteins onto their energy landscape framework? Section 3.3 provides some insight, but this needs to be stated more clearly.

6. In figure 3C, explain the trend between the blue curves. Why is it non-monotonic compared to the periodicity?

Minor remarks:

7. Figures are not referenced in the order they appear. E.g., Fig. 4 in section 2 before mentioning Fig 3.

8. The chain length of 200 does not mimic condensate-forming proteins. (Line 100). This is mimicking LCDs of condensate-forming proteins.

9. ‘mere shuffling’ what does this mean? What is the procedure for generating the randomized sequences.

10. Some places use ‘sticker spacers’ what is this? This phrasing is confusing/dilutes their message.

Reviewer #2: In this work, the authors present Alphabet-Free Energy Landscape framework of Stickers and Spacers (AFELAS) in which protein sequence is defined by specifying the relative “sticker” affinity of residues with respect to baseline “spacer” affinity. This study investigates how protein sequence patterns influence the material properties of biomolecular condensates through a so-called “energy landscape” framework. By using molecular dynamics simulations of a simple spring-bead polymer model, the authors show that periodic sticker patterns promote the formation of elasticity-dominated, stable condensates, while random patterns lead to viscosity-dominated, less stable condensates.

I LOVE this work. The authors thoroughly investigated the equilibrium and kinetic material properties with appropriate analysis methods. They compare their results with available experimental data (although there are very few, unfortunately). The structure of the manuscript is clear and concise. Their findings underscore the importance of sequence periodicity in determining condensate stability and fluidity, providing valuable insights into protein aggregation-related diseases and the design of synthetic biomaterials. I would strongly recommend my students to study this work as a role model. I just provide several comments below, which may help the authors improve the manuscript. The list is somewhat lengthy, but the comments are mostly editorial.

Major comments

1. The authors call their approach the energy landscape framework, but I don’t fully grasp the meaning. In the field of protein folding, the energy landscape is a well-defined object, where energy is represented as a function of reaction coordinates. In this work, the authors seem to use different sequence patterns as the “reaction coordinate,” but then, what is a quantity corresponding to energy? If it is not precisely defined, I am not sure if you can call it the “energy landscape” framework.

2. Since the main tool of this work is molecular dynamics simulations, I think that the authors should make sure if the simulated systems, particularly those for equilibrium properties, are fully converged. Can the authors add some discussion on this?

3. The authors fixed the chain lengths at Np=200, as exploring a distribution of chain lengths could improve the robustness of their findings. What is the main reason for choosing that? If there is any study published, please provide a reference.

4. This model assumes constant interaction strengths for stickers, which might not capture context-dependent interactions such as pH or ionic strength variations in biological environments. Can the authors comment on how to incorporate such effects in this model?

5. The authors sometimes introduce symbols and abbreviations without their definitions. For example:

5-a. In Fig. 1, what is P(δE)?

5-b. In line 75, what is WCA? It is used again in line 81.

5-c. In Fig. S1, the authors used abbreviations like PDSSM and RDSSM, which are never defined anywhere in the main text or the SI.

6. In the caption of Fig. 1, the authors mention “motif IDs.” How do the authors define the distinct motifs and their lengths?

7. In the caption of Fig. 2, the authors mention “The interaction strengths between distinct stickers are illustrated in the energy landscape diagram.” Which panel illustrates it?

8. Can the authors explain or provide a rationale behind the statement (in line 109), “Intriguingly, regardless of the permutation of the periodically distributed chain, the phase diagram of the randomly arranged sequences manifests approximately the same critical temperatures…”?

9. I wonder why the authors use S(q) instead of g(r). Both contain the same information, but for the readers who are not familiar with the reciprocal space representation, g(r) may be easier to understand. Do the authors have specific reasons for using S(q)?

10. In Fig. 3C, the behaviors of the Pn systems require some explanation. The curves cross each other, and there is apparently no trend along increasing n. I am a little worried if this is due to insufficient sampling.

11. In section 3.3 or Fig. 5, the details of the experimental systems (such as protein names) should be provided.

12. In Fig. 6B and 6D, why do we have “bumps” (significant deviations from the exponential decay) in the spacer distributions?

13. In Fig. 6F, it seems that the authors use the same curve for all three different systems. However, even when the three curves overlap, fluctuations may give some deviations. Hence, I recommend the authors use different colors for different curves.

14. The caption for Fig 7 seems to be truncated.

15. In lines 290-291, what do the authors mean by “randomly distributed frustrated sticker-spacer condensed phase”?

16. In lines 294-295, the authors claim that “randomly arranged stickers remain Gaussian in nature,” but is it true? Fig. 7F seems to indicate that randomly arranged stickers deviate from the Gaussian behaviour as time progresses.

17. In line 317, the authors state “However, Rgii degenerates as density increases.” What do they mean by “degenerates”?

18. In line 356, the authors claim that they continuously “evolve” sequence types. I believe that the authors mean “change.” I strongly recommend replacing the word “evolve,” as in the field of protein science, it is usually related to biological evolution.

Minor Comments

1. In line 15, is “membrane organless” a typo for “membraneless organelles”?

2. In line 23, is “sequence-condensate thermodynamics” a typo for “sequence-dependent thermodynamics”?

3. In line 33, can the authors provide some references for “evolutionary pressure optimizes the energy gap between unfolded and folded states at physiologic conditions to avoid deep kinetic traps thereby ensuring rapid and robust folding”?

4. In Figs. 2, 4 and S1, the texts are too small, and the images also seem squeezed. Can authors make it more readable?

5. In line 201, 211, 330, is “micell-like” a typo for “micelle-like”?

6. In lines 314-315, the phrase “where for chain lengths N p = 25, 50, 100 (Fig. 8 G).” is ungrammatical.

7. In the first row of Fig. 9, the snapshots are overlapped with the panel labels and the texts. Can the authors rearrange them to show more clearly?

8. In line 329, “Whereas cos−1 delineates the transition between oblate and prolate shapes.” is a fragmented clause.

9. In line 330, there is a comma before the colon. Is it a typo?

10. In line 374, the hyphen in “The van-Hove analysis” should be removed.

11. In line 378, is “micelles-like” a typo for “micelle-like”?

12. In the SI, the first line repeats word “map” twice.

**Have the authors made all data and (if applicable) computational code underlying the findings in their manuscript fully available?**

Reviewer #1: **No: **no code/availability statement

Reviewer #2: **No: **They only provide the analyzed data, which is not sufficient to reproduce and check their simulations.

PLOS authors have the option to publish the peer review history of their article (what does this mean?). If published, this will include your full peer review and any attached files.

Reviewer #1: No

Reviewer #2: No

**Figure resubmission:**While revising your submission, please upload your figure files to the Preflight Analysis and Conversion Engine (PACE) digital diagnostic tool, https://pacev2.apexcovantage.com/. PACE helps ensure that figures meet PLOS requirements. To use PACE, you must first register as a user. Registration is free. Then, login and navigate to the UPLOAD tab, where you will find detailed instructions on how to use the tool. If you encounter any issues or have any questions when using PACE, please email PLOS at figures@plos.org. Please note that Supporting Information files do not need this step. If there are other versions of figure files still present in your submission file inventory at resubmission, please replace them with the PACE-processed versions. **Reproducibility:**To enhance the reproducibility of your results, we recommend that authors of applicable studies deposit laboratory protocols in protocols.io, where a protocol can be assigned its own identifier (DOI) such that it can be cited independently in the future. Additionally, PLOS ONE offers an option to publish peer-reviewed clinical study protocols. Read more information on sharing protocols at https://plos.org/protocols?utm_medium=editorial-email&utm_source=authorletters&utm_campaign=protocols

---

## [Editor Report · Decision Letter 1]

23 Jan 2025

Dear Prof Potoyan,

We are pleased to inform you that your manuscript 'Decoding Biomolecular Condensate Dynamics: An Energy Landscape Approach' has been provisionally accepted for publication in PLOS Computational Biology.

We appreciate your thoughtful response to the reviewer comments and believe that the scholarly dialogue has improved the manuscript.

Best regards,

Peter M Kasson

Academic Editor

PLOS Computational Biology

Arne Elofsson

Section Editor

PLOS Computational Biology

---

## [Editor Report · Acceptance letter]

PCOMPBIOL-D-24-01736R1

Decoding Biomolecular Condensate Dynamics: An Energy Landscape Approach

Dear Dr Potoyan,

I am pleased to inform you that your manuscript has been formally accepted for publication in PLOS Computational Biology. Your manuscript is now with our production department and you will be notified of the publication date in due course.

With kind regards,

Zsofia Freund
